

# A prospective randomized trial examining health care utilization in individuals using multiple smartphone-enabled biosensors

Cinnamon S. Bloss[1],[*], Nathan E. Wineinger[1],[*], Melissa Peters[1], Debra L. Boeldt[1], Lauren Ariniello[1], Ju Young Kim[2], Judith Sheard[1], Ravi Komatireddy[1], Paddy Barrett[1] and Eric J. Topol[1],[3],[4]

[1] Scripps Translational Science Institute, La Jolla, California, United States
[2] Department of Family Medicine, Seoul National University Bundang Hospital, Seoul, South Korea
[3] Department of Molecular and Experimental Medicine, The Scripps Research Institute, La Jolla, United States
[4] Division of Cardiovascular Diseases, Scripps Health, San Diego
[*] These authors contributed equally to this work.

Corresponding author
Eric J. Topol, etopol@scripps.edu

## ABSTRACT

**Background.** Mobile health and digital medicine technologies are becoming increasingly used by individuals with common, chronic diseases to monitor their health. Numerous devices, sensors, and apps are available to patients and consumers–some of which have been shown to lead to improved health management and health outcomes. However, no randomized controlled trials have been conducted which examine health care costs, and most have failed to provide study participants with a truly comprehensive monitoring system. **Methods.** We conducted a prospective randomized controlled trial of adults who had submitted a 2012 health insurance claim associated with hypertension, diabetes, and/or cardiac arrhythmia. The intervention involved receipt of one or more mobile devices that corresponded to their condition(s) (hypertension: Withings Blood Pressure Monitor; diabetes: Sanofi iBGStar Blood Glucose Meter; arrhythmia: AliveCor Mobile ECG) and an iPhone with linked tracking applications for a period of 6 months; the control group received a standard disease management program. Moreover, intervention study participants received access to an online health management system which provided participants detailed device tracking information over the course of the study. This was a monitoring system designed by leveraging collaborations with device manufacturers, a connected health leader, health care provider, and employee wellness program–making it both unique and inclusive. We hypothesized that health resource utilization with respect to health insurance claims may be influenced by the monitoring intervention. We also examined health-self management. **Results & Conclusions.** There was little evidence of differences in health care costs or utilization as a result of the intervention. Furthermore, we found evidence that the control and intervention groups were equivalent with respect to most health care utilization outcomes. This result suggests there are not large short-term increases or decreases in health care costs or utilization associated with monitoring chronic health conditions using mobile health or digital

medicine technologies. Among secondary outcomes there was some evidence of improvement in health self-management which was characterized by a decrease in the propensity to view health status as due to chance factors in the intervention group.

## INTRODUCTION

Hypertension, diabetes mellitus, and cardiac arrhythmias are chronic diseases with a significant health burden. The high prevalence and well-characterized complications of these conditions result in negative impacts to quality of life, morbidity, and mortality. Vast medical, scientific, and engineering resources have been devoted in efforts to find ways to improve diagnosis, treatment, management, and prevention, including advances in diagnostic technology (*Willits et al., 2014*; *Zhang et al., 2014*; *National Institute for Health and Care Excellence, 2015*), dissemination of identifiable risk factors (*Wilson et al., 1998*; *Chobanian et al., 2003*; *American Diabetes Association, 2014*), and development of pharmaceuticals (*The ALLHAT Officers and Coordinators for the ALLHAT Collaborative Research Group, 2000*; *The ALLHAT Officers, 2002*; *Uzu et al., 2005*; *Ernst et al., 2006*). Nevertheless, the continued maintenance of these efforts and the costs associated with direct patient care of individuals with these conditions remains a sizable fraction of health care costs (*Kim et al., 2011*; *Davis, 2013*; *Yang et al., 2013*).

Despite these efforts, the management of these conditions remains challenging (*Hansen et al., 2005*). Patient engagement, medication adherence, and adherence to treatment strategies is variable and often suspect (*Guyatt et al., 1986*; *Hansen et al., 2005*). Furthermore, poor communication between patients and their health care providers can accentuate these issues. The end result is often a major clinical decompensation event that could have been largely preventable. In addition to these clinical consequences, these challenges also result in economic consequences such as high utilization of inpatient resources and emergency departments, as well as readmissions (*Sander & Giles, 2011*). One would surmise that better, more informed management of disease would lead to better long-term health outcomes and thereby lower health care resource utilization.

Understanding both this problem of day-to-day poor health care management and ubiquity of smartphones and other mobile computing platforms in our daily lives, numerous device manufacturers have developed biomedical sensors designed for patient consumers which measure meaningful physiologic metrics (*National Institute for Health and Care Excellence, 2015*). These sensors often utilize a smartphone to display information, while some can employ the internet network connectivity of the smartphone to transmit data to remote servers for additional display, storage, or analytics. Individuals appropriately using such devices can monitor their condition in their own real world setting–potentially making management of disease more personalized and more engaging.

This data may provide individuals with early recognition of disease symptoms and consequences of behaviors, which can allow patients and providers to make proactive health care decisions, thereby leading to improved health outcomes and potentially reduce health care utilization. However, there is potential that such monitoring may lead to a short-term increase in health care resource utilization even if appropriate, or over-utilization while patients are learning to recognize which readings constitute normal variation and which readings indicate a health issue requiring medical attention.

In large part, the jury is still out if chronic disease monitoring using mobile health and digital medicine technology will, on its own, improve health outcomes (*Steinhubl, Muse & Topol, 2013*; *Steinhubl, Muse & Topol, 2015*). Many previous studies have shown improvements, but many others have shown none (*Free et al., 2013a*; *Free et al., 2013b*; *Hamine et al., 2015*; *Karhula et al., 2015*). Whether or not a patient has improved health as a result such monitoring likely depends on the behavior of the individual and the technology itself. Motivated individuals using an informative device which captures actionable data are likely to see improvements, while unmotivated individuals using devices which capture meaningless or nonactionable information will see no benefit. What has yet to be fully explored is how such monitoring will impact health care utilization if digital medicine technology becomes embraced by the medical establishment in the face of an increasingly informed, technology-embracing patient population (*Boeldt et al., 2015*).

In light of this, we conducted a prospective, randomized-controlled trial designed to assess the impact of mobile health monitoring on short-term health care utilization in individuals with hypertension, diabetes, or arrhythmia. The intervention consisted of a comprehensive, integrated monitoring system that included wireless medical devices designed to be used with a smartphone, a smartphone with appropriate monitoring applications, a web-based care management portal and iOS-based mobile application where patients could access their data, and a staff of nurses and technicians available for assistance. Given the potential for both short-term and long-term impacts on health care utilization, we hypothesized that the intervention may impact health care costs compared to standard disease management practices. We also examined how this intervention influenced health self-management. We conducted this mobile health management strategy on 160 employees and dependents from a large health care system.

## MATERIAL AND METHODS

### Study design

The research design was a prospective, simple randomized controlled, two-group, pre-post intervention trial. Of 21,691 individuals insured by Scripps Health (employees and dependents) and who submitted at least one claim in 2012, 3,998 individuals age 18 or over who had billed a claim with a current procedural terminology (CPT) code related to hypertension, insulin-dependent or non-insulin dependent diabetes, and/or with arrhythmia were identified and, if eligible, offered study participation. No stratification with respect to condition was employed. Additional inclusion and exclusion criteria are included in Supplemental Methods. The study period was six months and the trial took

place between July 2013 and December 2014. The study was approved by the Scripps Institutional Review Board (approval # IRB-12-6019).

## Study recruitment

The 3,998 eligible individuals were ranked according to the amount of their 2012 health insurance claims billed for the three study conditions. Recruitment proceeded in blocks starting with individuals in the highest 25% in terms of dollar amount, then the highest 50% and so on. For each block of individuals, a letter was sent describing the trial and disease management program. Within two to four weeks, the letter was followed up with a telephone call made by a HealthComp nursing staff member in which the study was explained. HealthComp is the third party administrator for Scripps Health. A maximum of three calls were attempted before a prospective participant was considered not reachable. For those prospective participants who expressed interest, a link to the online informed consent was sent via email. Prospective participants were asked to read and sign the online consent. Once consented, participants were directed to complete an online baseline survey. Afterwards, participants were randomized to control or intervention and brought in for an enrollment visit with an unblinded research coordinator. The participants were blind to their assigned group prior to enrollment. It was explained to all participants that their employer would not have access to any of their medical information used for the study.

## Study enrollment

At the enrollment study visit, individuals assigned to the intervention arm were provided with a study iPhone 4 or 4s (even if they owned one) and one or more mobile devices that corresponded to their condition(s): a Withings Blood Pressure Monitor (hypertension), Sanofi iGBStar Blood Glucose Meter (diabetes), or AliveCor Mobile ECG (arrhythmia). As part of the intervention, participants were also supplied with an online account to HealthyCircles™. HealthyCircles is a Qualcomm Life health care coordination and management platform with an integrated suite of management and consumer portals that can deliver chronic disease education and connect users to their families, caregivers, and health care professionals. As part of the study, HealthComp nursing staff had access to the HealthyCircles care management dashboard which displayed the participant's device monitoring results and trends over time. Device readings collected by the participant were wirelessly uploaded to the patient's HealthyCircles account and made available to the HealthComp nurses as well as the patient via the study phone or a computer. Example displays are included in Figs. S1–S3. Also included in the management platform were reminders for monitoring, information about the participant's disease condition, and general health behavior recommendations. Participants randomized to the intervention group were trained on how to use their phone, the HealthyCircles mobile applications and portal, and their medical device(s). All study participants, including participants randomized to the control arm, were enrolled in the HealthComp disease management program, which involved outreach by HealthComp nursing staff for purposes of relaying medical education and wellness information with regard to disease prevention and

chronic disease management. In addition, systolic and diastolic blood pressure readings were recorded at baseline and follow-up on individuals with hypertension, whereas blood glucose and hemoglobin A1c levels were recorded on diabetics. Participants were also provided with a contact email and phone number they could use to reach a study staff member for technical and other study support.

Participants in the monitoring group were asked to take readings as follows: hypertension: twice per day, three days per week, first one in the morning; insulin-dependent diabetes: three times per day, once before each meal, and once before bed every day; non-insulin dependent diabetes: once per day before meal, three times per week; and arrhythmia: when symptomatic (Table S4). If their monitoring fell below the level defined in the "Poor Compliance" range, the HealthComp nurse would send a secure email through the HealthyCircles message center reminding them of the monitoring schedule. They were also asked some compliance-related questions and provided with strategies for getting back on schedule with the program. If the participant's physician recommended a monitoring schedule that was more frequent than that required for the study, participants were encouraged to follow their physician's instructions. Also, if participants experienced other symptoms, for example hypertension: visual changes, "bounding" pulse, chest discomfort, nausea; diabetes: fatigue, visual changes, pre-syncopal symptoms, dyspnea, nausea, vomiting; arrhythmia: chest discomfort, palpitations, rapid heart rate, feeling of "skipped beats," dyspnea, nausea, pre-syncopal symptoms they were recommended to take additional measurements. Individuals were instructed on how to navigate the online disease management program at http://connect.healthcomp.com.

### Study procedures

Study participants were asked to attend both an enrollment study visit (baseline) and end-of-study visit (follow-up) six months later, and to complete both a baseline and follow-up survey on SurveyMonkey. At the mid-point of the study (i.e. three months), each participant also received an email seeking any feedback about their experience thus far, or asking if they were having any problems or had any questions.

### Outcome measures

For both the treatment and control groups, outcomes were assessed using claims data during the enrollment and termination visits. Primary outcomes were health care resource utilization as measured by health insurance claims and visits to the hospital during the study period (details below). Secondary outcomes were health self-management as indicated by validated measures of health locus of control (*Wallston, Stein & Smith, 1994*), health self-efficacy (*Lorig et al., 1989*), and patient activation (*Hibbard et al., 2005*).

### Health insurance claims

All health insurance claims from January 1, 2013 through December 31, 2014 were collected on each study participant. For each individual, the total claims, condition-specific claims, pharmaceutical claims amounts (all in dollars) were calculated.

Condition-specific claims were the total amount in claims related to one of the three study conditions monitored. For example, the amount of hypertension claims was the amount in claims in dollars with an ICD-9 code for hypertension. Totals were calculated for a period of 6 months prior to study enrollment (baseline claims; Table S5) and 6 months during enrollment (enrollment claims; Table S6). Claims were further partitioned into four categories: office visits, emergency room visits, inpatient stays, and all visits (all in number of occurrences). Differences in enrollment claims were then compared to baseline claims between the control and monitoring groups, as well as between groups with specific conditions. Data are available in Supplemental Data.

### Health self-management

Information on health self-management was collected through the baseline and follow-up survey. The outcomes of interest were: 1) the four subscales of health locus of control (Internal, Chance, Doctor, Others) as assessed by the Form C of the Multidimensional Health Locus of Control (MHLC) 18-item scale (*Wallston, Stein & Smith, 1994*); 2) health self-efficacy as assessed by the Stanford Patient Education Research Center (PERC) 6-item scale (*Lorig et al., 1989*); and 3) patient activation using the Patient Activation Measure 13-item measure (*Hibbard et al., 2005*). Health locus of control measures an individual's belief that his or her health is dependent on factors within his or her control; health self-efficacy measures an individual's confidence managing his or her health; and patient activation measures an individual's skills and knowledge in managing his or her own health.

### Device usage

Device usage statistics were recorded for each study participant in the monitoring group. Whenever a study participant used a device, the time, date, and user information of that particular reading was sent to a database managed by Qualcomm and available to participants through Healthy Circles. In the case of the Withings Blood Pressure Monitor and IBG Star, the reading measurements (i.e. blood pressure and blood glucose levels) were also recorded; while PDFs corresponding to the AliveCor arrhythmia assessments were saved. An issue was encountered where a subset of Withings measures could not be accurately determined. These measures were omitted. There were 21 study participants affected in varying severity, 10 of which had this issue present in all data and 8 others had this issue present in at least 16% of the data (the other two 6% and 0.5%). As accurate device usage information on these 18 individuals could not be determined, it was treated as missing. For all other study participants, the total number of readings taken on each device was recorded.

### Statistical analyses and sample size justification

Between group differences were compared using a paired two sample t-test or Mann-Whitney test in cases of small sample sizes and skewed outcomes (e.g. health insurance claims). By using this paired approach we better model the change in outcomes of interest induced by the intervention and reduce the influence of baseline confounders in the

association statistics. Equivalence testing was performed using the two one-sided test for equivalence using a magnitude of region of similarity equal to half a standard deviation for each outcome. The study was designed to be powered (*a priori*) to detect a one office visit difference between the control and monitoring arm (assuming a standard deviation of two office visits).

## RESULTS

### Participant demographics and information

Study participant demographics are presented in Table 1. Participants in the control and monitoring groups were roughly equivalent with respect to common demographics and disease, which is consistent with the randomization process. A total of 89 had only hypertension, 9 non-insulin dependent diabetes, 6 arrhythmia, 5 insulin-dependent diabetes, and 51 with more than one of these conditions. The study enrollment flow chart is presented in Fig. S7. Of the 160 individuals enrolled in the study, 130 completed both the baseline and follow-up assessments (n = 65 control, n = 65 monitoring; p = 0.14). Using Google Analytics we observed a total of 3,670 sessions (after quality control filtering) to the HealthyCircles online disease management program over the course of the study (Fig. S8), with 7.17 page visits per session, and average session duration of 11 minutes and 18 seconds. Google Analytics does not provide easily accessible individual user website traffic data. We assessed weekly compliance of the intervention in the monitoring group based on device usage (e.g., an individual with hypertension would be compliant in a given week if they used the device at least six times that week). We observed compliance rates were largely uniform (mean = 50%), with 66% of individuals deemed compliant at least one-third of the weeks.

### Health insurance claims

Health insurance claims during the period of 6 months prior to study enrollment did not differ between control and monitoring groups (Table S5). The average total amount of health insurance claims during this period was $5,712 (sd = $19,234; median = $976), and we observed no difference in claims between individuals with different disease conditions (p = 0.99). The average number of office visits was 4.1 (sd = 4.2; median = 3); the average number of emergency room visits was 0.10 (sd = 0.45; median = 0); and the average number of inpatient stays was 0.53 (sd = 3.10; median = 0). None of these claim categories differed statistically between conditions.

We did not observe any differences in health insurance claims between control and monitoring groups during the 6 months of study enrollment (Table S6). This trend also persisted when we accounted for baseline claims (Table 2). The average total amount of health insurance claims in the monitoring group was $6,026 while the average amount in the control group was $5,596 (p = 0.62). We note these averages are consistent with average total amount in health insurance claims across the entire sampling frame (mean = $5,305), indicating that health insurance claims in the monitoring group were not grossly different from the average patient (i.e., individuals not enrolled in the study).

**Table 1 Study participant demographics.** Values are in counts, proportions in parentheses (proportions) unless otherwise noted.

| | Monitoring | Control | p-value |
|---|---|---|---|
| N (# completed) | 75 (65) | 85 (65) | 0.47 |
| Hypertension | 67 (89) | 71 (84) | 0.29 |
| NIDDM | 10 (13) | 17 (20) | 0.26 |
| IDDM | 10 (13) | 10 (12) | 0.76 |
| Arrhythmia | 10 (13) | 19 (22) | 0.14 |
| Comorbidity | 21 (28) | 30 (35) | 0.41 |
| Gender (% Female) | 50 (67) | 62 (73) | 0.24 |
| Age, Mean (SD) | 56 (9.0) | 55 (9.8) | 0.45 |
| Ethnicity, Caucasian | 57 (76) | 62 (73) | 0.39 |
| Education | | | 0.25 |
| High School or Less | 10 (13) | 19 (22) | |
| College | 32 (43) | 37 (44) | |
| More than College | 33 (44) | 29 (34) | |
| Family Size | | | 0.87 |
| Single | 12 (16) | 13 (15) | |
| Two | 27 (36) | 34 (40) | |
| Three or More | 36 (48) | 38 (45) | |
| Income | | | 0.09 |
| < $50,000 | 10 (13) | 11 (13) | |
| $50k–$149k | 47 (63) | 58 (68) | |
| >$149k | 18 (24) | 16 (19) | |
| Current Non-Smoker | 45 (60) | 64 (75) | 0.04 |
| Alcohol Use, <1/week | 54 (72) | 65 (77) | 0.31 |
| Active Exerciser | 37 (49) | 37 (44) | 0.46 |
| Smartphone owned | | | 0.76 |
| Did not own | 11 (17) | 10 (15) | |
| Owned non-iPhone | 20 (31) | 24 (37) | |
| Owned iPhone | 34 (52) | 31 (48) | |

We also did not observe any differences between the groups with respect to office visits ($p = 0.46$), inpatient stays ($p = 0.82$), emergency room visits ($p = 0.06$), or pharmacy claims ($p = 0.60$). The total health insurance claims amount during enrollment also did not differ by condition ($p = 0.50$), and we similarly observed no differences in claims specific to each condition or multiple conditions (Table S6).

Alternatively, we examined the differences in health care utilization using an equivalence testing approach. Using a magnitude of region of similarity equal to half a standard deviation for each outcome, in general we discovered that health care utilization was roughly equivalent between groups (Table 2). We discovered that monitoring and control groups were roughly equal with respect to total health insurance claims dollars ($p = 0.027$), pharmacy claims ($p = 0.037$), office visits ($p = 0.038$), inpatient stays ($p = 0.042$), and total hospital visits ($p = 0.014$). This suggests that there is unlikely to be

**Table 2 Health care utilization outcomes.** Top: mean (standard deviation); bottom: median (IQR). $P_{Diff}$, p-value testing difference between control and monitoring group; $P_{Equiv}$, p-value testing equivalence between groups; *, Median and IQR all zero.

| | Baseline | | Follow-up | | Mean Difference | | | |
| --- | --- | --- | --- | --- | --- | --- | --- | --- |
| | Control N = 85 | Monitoring N = 75 | Control N = 65 | Monitoring N = 65 | Control N = 65 | Monitoring N = 65 | $P_{Diff}$ | $P_{Equiv}$ |
| Total Claims ($) | 4,265 (10,190) | 7,159 (25,251) | 5,596 (22,187) | 6,026 (21,426) | 1,331 (21,042) | −1,133 (31,465) | 0.62 | 0.027 |
| | 961 (3,166) | 990 (2,340) | 807 (2,734) | 845 (2,273) | 0 (2,372) | 0 (1,780) | | |
| Condition Claims ($) | 1,512 (6,868) | 2,434 (14,296) | 6,165 (37,153) | 630 (21,43) | 4,653 (35,795) | −1,805 (14,406) | 0.50 | 0.105 |
| | 163 (375) | 117 (387) | 111 (379) | 179 (516) | 0 (208) | 0 (283) | | |
| Pharmacy Claims ($) | 1,519 (2,687) | 1,859 (5,315) | 1,667 (2,780) | 2,188 (6,340) | 147 (1,057) | 329 (1,860) | 0.60 | 0.037 |
| | 325 (1,590) | 345 (1,164) | 611 (1,603) | 340 (1,458) | 11 (531) | 0 (321) | | |
| Total Visits (#) | 4.49 (5.01) | 4.92 (6.51) | 4.17 (4.21) | 4.77 (5.35) | −0.32 (3.75) | −0.15 (6.35) | 0.57 | 0.014 |
| | 3 (6) | 3 (4) | 2 (7) | 3 (5) | 0 (2) | 0 (3) | | |
| Office Visits (#) | 4.11 (4.41) | 4.05 (4.09) | 3.95 (3.92) | 4.32 (4.48) | −0.15 (3.30) | 0.28 (3.60) | 0.46 | 0.038 |
| | 3 (5) | 3 (4) | 2 (5) | 3 (4) | 0 (2) | 0 (2) | | |
| ER Visits (#)* | 0.17 (0.60) | 0.03 (0.17) | 0.05 (0.37) | 0.06 (0.30) | −0.12 (0.72) | 0.03 (0.35) | 0.06 | 0.137 |
| Inpatient Stays (#)* | 0.22 (0.94) | 0.85 (4.27) | 0.17 (0.89) | 0.38 (1.88) | −0.05 (1.16) | −0.46 (4.30) | 0.82 | 0.042 |

substantial short-term changes in health care utilization as a result of the monitoring intervention.

We also examined health insurance utilization in a subset of the monitoring group who we were able to assess as being compliant with the study protocol in at least one-third of the weeks of the study. Again, we did not observe any differences with respect to the total amount of health insurance claims (p = 0.17), office visits (p = 0.34), or inpatient stays (p = 0.34). Though there was slight trend towards an increase in emergency room visits among these participants in the monitoring group (mean increase = 0.10) compared to the controls (mean decrease = 0.12; p = 0.027).

## Health self-management

Additionally, we examined the relationship between monitoring/control group assignment and health self-management using baseline and follow-up survey responses. We quantified differences in measures of health locus of control, self-efficacy, and patient activation (Table 3). Each of these are validated measures designed to address how an individual perceives his or her health and health management. We did not find differences in changes in self-efficacy (p = 0.85) or patient activation (p = 0.68) between groups. In both cases, both the control and monitoring groups did not differ between baseline and follow-up. The average Stanford Patient Education Research Center (PERC) 6-item self-efficacy scale was 7.9 and 8.0 across both groups at baseline and follow-up, respectively. Meanwhile, the average Patient Activation Measure 13-item measure was 73 and 76 across both groups at baseline and follow-up, respectively. However, one component of Form C of the Multidimensional Health Locus of Control (MHLC) 18-item scale, the propensity to view health status as due to chance factors (MHLC Chance), showed improvement in the intervention group as compared to controls

**Table 3 Mean values of health self-management outcomes of study.** Standard deviation in parentheses.

| | Baseline | | Follow-up | | Mean Difference | | | |
| | Control<br>N = 85 | Monitoring<br>N = 75 | Control<br>N = 65 | Monitoring<br>N = 65 | Control<br>N = 65 | Monitoring<br>N = 65 | Effect<br>Size | p |
|---|---|---|---|---|---|---|---|---|
| MHLC Internal | 26.0 (6.0) | 26.1 (6.7) | 26.3 (6.0) | 26.1 (5.9) | 0.08 (6.4) | 0.34 (5.3) | 0.11 | 0.80 |
| MHLC Chance | 12.3 (5.9) | 12.3 (5.6) | 13.4 (5.8) | 11.3 (5.3) | 1.30 (5.0) | −0.76 (4.9) | −0.93 | 0.02 |
| MHLC Doctor | 14.9 (2.7) | 15.3 (2.6) | 14.8 (3.0) | 15.7 (2.3) | −0.22 (3.8) | 0.43 (2.5) | 0.37 | 0.34 |
| MHLC Others | 8.4 (3.6) | 7.6 (3.0) | 8.1 (3.3) | 7.9 (3.1) | −0.15 (3.8) | 0.50 (3.2) | 0.35 | 0.59 |
| PERC Self-Efficacy | 7.5 (2.0) | 8.4 (1.4) | 7.8 (1.7) | 8.4 (1.7) | 0.31 (2.1) | −0.05 (1.4) | −0.27 | 0.85 |
| Patient Activation | 70.2 (14.2) | 77.6 (13.1) | 74.6 (18.9) | 79.0 (20.9) | 4.35 (18.2) | 0.75 (18.4) | −0.84 | 0.68 |

**Abbreviations:**
MHLC, Multidimensional Health Locus of Control; PERC, Patient Education Research Center.

($\Delta$ = 2.06; p = 0.020). We simultaneously observed an approximately 1.3 increase in the scale in the control arm and 0.8 decrease in the intervention arm. Thus, compared to controls, participants in the intervention arm were less likely to view their health status as due to chance. We did not observe any group differences with respect to the other health locus of control components. In each group, the average scores at the follow-up visit were within 0.5 of the baseline scores (Table 3).

Among the 138 individuals who enrolled in the study with a prior indication of hypertension, we obtained both baseline and follow-up systolic and diastolic blood pressure readings on 112 participants (n = 61 monitoring; n = 61 control). The average systolic blood pressure did not differ between baseline and end-of-study in the monitoring group (p = 0.32), control group (p = 0.12), or between groups (p = 0.56). However, the average diastolic blood pressure dropped 3.6 mmHg in the monitoring group (p = 0.035) and 6.1 mmHg in the control group (p = 0.0036); though again there was no difference between groups (p = 0.35). Likewise, among the 47 individuals who enrolled in the study with a prior indication of diabetes, we obtained hemoglobin A1c levels on 31 participants at both time points. However, hemoglobin A1c levels did not differ between baseline and the end-of-study within each group or between groups (p = 0.98).

## Device usage

Study participants in the monitoring group who completed the follow-up study visit used one of the monitoring devices a total of 10,305 times (Fig. S9). This includes 6,356 blood pressure readings, 3,440 blood glucose readings, and 509 arrhythmia readings. The average number of blood pressure readings was 151 (sd = 84; median = 154) with a maximum of 436. Four of 42 (10%) study participants had fewer than three times measurements over the course of the study. All others had more than 60 measurements. The average number of blood glucose readings was 248 (sd = 268; median = 125). Four of 14 (29%) study participants did not record a reading. Meanwhile, the average number of arrhythmia readings was 57 (sd = 54; median = 53) with one individual of nine (11%) not using the device.

## DISCUSSION

Our study constitutes a major advancement over existing studies that have examined mobile health technologies by virtue of its design features. First we deployed a gold-standard prospective, randomized design with an intervention that included multiple key components relevant for management of three chronic conditions with high morbidity and mortality. This intervention included the use of three state-of-the-field wireless smartphone-enabled remote monitoring medical devices. Furthermore, data from the devices was aggregated using the Qualcomm Life cloud-to-cloud data integration capability. Data visualization was then provided to study participants through an online care coordination application where participants could view their device readings through web and mobile mediums throughout the course of the intervention period. Thus, we feel that compared to previous studies in the mobile health space, our intervention more closely mirrors a future in which chronic disease monitoring using mobile biomedical sensors is embraced by the health care community. This requires a system that brings together device manufacturers, mobile health telecommunication expertise, health care providers, and employee wellness programs–all of which we utilized in the development and implementation our mobile health monitoring intervention.

We enrolled 160 study participants in the study, achieving low drop-out particularly in the monitoring group where 87% of participants completed all aspects of the study. We also had relatively good compliance among individuals in the monitoring group. Hypertensive study participants on average recorded one blood pressure measurement per day, roughly what we requested (twice per day, three days per week; 6 total per week). In total, individuals in the monitoring group provided over 9,000 blood pressure, blood glucose, and electrocardiogram readings which we will are now examining for interesting trends in the entirety of data we collected. Interestingly, these individuals used the HealthyCircles online disease management program somewhat sparingly (3,670 uses over six months for 65 study participants). Instead, many users preferred the mobile tracking applications. Future research could explore different means of providing data back to individuals monitoring their condition.

Overall we found little in terms of differences in health insurance claims between individuals enrolled in the control and monitoring arm. This is significant because we were powered to detect a moderate difference–approximately a doubling of health insurance claims dollars. This suggests that while there may be small short-term increases in health care utilization as a result of mobile health monitoring, there is likely not a major effect. Our equivalence testing results reiterated this finding. We also expect that any short-term effect would decrease over time as a user's comfort with monitoring and understanding of their data improves. Importantly, our six month study period fails to capture the potential competing long-term decrease in health care utilization that may occur as a result of monitoring leading to improved health management and health outcomes. Taken collectively, we feel any apprehension directed at consumer mobile health monitoring with respect to over-utilization of health care resources should be tempered.

Meanwhile, we found some evidence of improved health self-management in individuals who received the intervention, which was characterized by a decrease in the propensity to view health status as due to chance factors. One possible explanation is that this shift was due to the ability to remotely, and at will, track personal biometric indices important for one's condition. Another explanation is that the actual information gleaned from the readings prompted the users to consider how they might make behavioral changes that would impact those metrics. Clarifying this mechanism of action could enable the development of future digital medicine interventions that are refined in such a way as to optimally impact health locus of control.

We encountered several challenges executing this project as a result of its complexity. One particular challenge was effectively dealing with the involvement of and collaboration between multiple entities, including industry, research, and clinical partners in the digital medicine space. Necessary legal agreements, data pipelines, and working arrangements were required to facilitate the study initiation and execution. In total, the study involved over five different Scripps departments, ten different companies, development and execution of at least eight different contracts or legal agreements, five different terms of use that a study participant could potentially have to agree to, and creation of six participant instruction or "set-up" guides. As this demonstrates, the conduct and deployment of digital medicine trials can present unique challenges that future work in this area could help address. We also encountered technological issues. Out of 75 individuals enrolled in the monitoring group, 21 (28%) experienced issues that required the research team to log at least one help desk ticket due to technical issues with the participant's phone, device(s), or connection to the online portal. Furthermore, 10 of these individuals had more than one help desk ticket submitted and at least 20 individuals had to have either the iPhone and/or the device replaced altogether. Technical issues are, of course, inevitable when pursuing innovative interventions that leverage new technologies. However, in order for such interventions to effectively provide benefit to the user they have to be seamless in order to minimize participant fatigue. Exposing study participants to such issues has the potential to create biases in the study results. Though we did not observed a difference in drop-out rate between individuals who submitted a help desk ticket and those that did not ($p = 0.27$), minimizing technological issues should be an important consideration in the design of future digital medicine trials.

In conclusion, we have presented the first prospective randomized trial of a digital medicine intervention with multiple smartphone-enabled biosensors, data aggregated and visualized through an online connected health platform, deployed with three high morbidity and mortality chronic diseases examining health care utilization. Our results suggest there is little to no short-term increase in health care utilization as a result of participation in a comprehensive mobile health monitoring care coordination platform. Meanwhile, we did see some improvement in health self-management. Future work should explore the potential reduction in long-term health care utilization as a result of potentially improved health management due to mobile health monitoring.

### Funding

This research is funded in part by a NIH/NCATS flagship Clinical and Translational Science Award Grant (1UL1 TR001114), Qualcomm Foundation Scripps Health Digital Medicine Research Grant, and Scripps Health's Division of Innovation and Human Capital and Division of Scripps Genomic Medicine. Support for the study is also provided by HealthComp Third Party Administrator, Sanofi, AliveCor, and Accenture. The funders had no role in study design, data collection and analysis, decision to publish, or preparation of the manuscript.

### Grant Disclosures

The following grant information was disclosed by the authors:
NIH/NCATS flagship Clinical and Translational Science Award Grant: 1UL1 TR001114.

### Competing Interests

Eric J Topol is an Academic Editor for PeerJ.

### Author Contributions

- Cinnamon S Bloss conceived and designed the experiments, performed the experiments, analyzed the data, wrote the paper, prepared figures and/or tables, reviewed drafts of the paper.
- Nathan E Wineinger conceived and designed the experiments, analyzed the data, wrote the paper, prepared figures and/or tables, reviewed drafts of the paper.
- Melissa Peters performed the experiments, reviewed drafts of the paper.
- Debra L Boeldt conceived and designed the experiments, reviewed drafts of the paper.
- Lauren Ariniello analyzed the data, reviewed drafts of the paper.
- Ju Young Kim analyzed the data, reviewed drafts of the paper.
- Judith Sheard reviewed drafts of the paper.
- Ravi Komatireddy conceived and designed the experiments, reviewed drafts of the paper.
- Paddy Barrett conceived and designed the experiments, reviewed drafts of the paper.
- Eric J Topol conceived and designed the experiments, wrote the paper, reviewed drafts of the paper.

### Human Ethics

The following information was supplied relating to ethical approvals (i.e., approving body and any reference numbers):
    The study was approved by the Scripps Institutional Review Board (approval # IRB-12-6019).
## Clinical Trial Ethics

The following information was supplied relating to ethical approvals (i.e., approving body and any reference numbers):

The study was approved by the Scripps Institutional Review Board (approval # IRB-12-6019).

## Data Deposition

Data can be found in the Supplemental Information.

## Clinical Trial Registration

The following information was supplied regarding Clinical Trial registration:

The study was registered at clinicaltrials.gov (Trial Registration Identifier # NCT01975428).

## Supplemental Information

Supplemental information for this article can be found online at http://dx.doi.org/10.7717/peerj.1554#supplemental-information.

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
