# Peer review of "A prospective randomized trial examining health care utilization in individuals using multiple smartphone-enabled biosensors"

_PeerJ, doi:10.7717/peerj.1554_

## Round 0.1 · original submission · Minor Revisions

· Academic Editor

Minor Revisions

First, sorry for the delay. I enjoyed reading this manuscript. Please see the thoughtful comments from the reviewers and edit/reply. In my opinion acceptance isn't mandating on editing for all of them, but I think the paper will be strengthened by considering them and either changing manuscript and/or replying.

·

Basic reporting

Bloss et al have written a very interesting analysis of a randomized controlled trial that investigated whether smartphone-enabled biosensors would lead to changes in health care utilization among individuals with hypertension and/or diabetes.

The reporting is sound. Because the biosensors were a critical part of the intervention, the authors may wish to mention the specific biosensors in the abstract.

Experimental design

The experimental design utilized a gold-standard RCT format.

Validity of the findings

The findings appear to be valid. A few issues which the authors may want to comment on:
--Was a health monitoring / disease management group an appropriate control group for this study? Would a usual care group have been more appropriate?
--What was the plausible mechanism by which biosensor monitoring was hypothesized to reduce health care utilization? How were the authors able to disprove this hypothesis? Did the authors think that biosensor monitoring would lead to changes in the way that participants managed their hypertension or diabetes at home, and that this would in turn reduce clinic, ED and hospital visits? If so, was this data captured - what self-management techniques participants used in response to their data?
--Were the health care utilization metrics chosen the right outcomes to use for this intervention? For example, did users with more vs less well managed hypertension or diabetes have differential rates of health care utilization? If so, can this data be presented? And if not, then does this undercut the argument that biosensor monitoring could have plausibly led to reductions in health care utilization?
--The compliance in this study was fairly low. One third of patients were largely non-compliant altogether, as only 2/3 were compliant for at least 1/3 of the weeks. What do the authors conclude from this? Do they think that more intensive coaching will lead to better compliance? Or a better communication of the value of monitoring at the beginning of the intervention? Or more attractive or easier to use apps?
--Did the authors collect symptom, physical function or HRQOL data? Were there any differences seen in these PROs between the intervention and control group?
--Were the participants familiar with or using any other self-tracking biometrics before the study (e.g. activity), and did this make them more or less likely to comply (or derive benefit from) the biosensors used in this study?
--Did the participants with the highest or lowest biosensor monitoring compliance show any health care utilization differences in comparison with the other participants?

Additional comments

My comments are included above. I think this is a very interesting study with high contemporary relevance, and will hopefully be one of several studies that will investigate the ability of self-monitoring biometrics to favorably impact health outcomes.

·

Basic reporting

No comments.

Experimental design

No comments.

Validity of the findings

Lines 280-284: Why was there so little usage of the online disease management program? 3670 over 6 months for 130 people (those interested enough to do both the baseline and follow-up) is low. Do you have a sense as to why this was the case? Also, was there no login information to map sessions back to users ids?

Lines 284-286: It is unclear exactly what constitutes compliance. Did the user have to take every required reading? Also was there any information gathered on the level of computer literacy/sense of self-efficacy of the users? This might have an impact on their interaction with the technology.

Lines 410-418: It seems that these technological issues encountered indicate there were some issued with the usability of the system. The authors dismiss these as “…inevitable when pursuing innovative interventions that leverage new technologies” – but these issues may have impacted the results. A detailed discussion of these issues and their potential impact would be appropriate.

Additional comments

One comment that doesn't fit in the above areas -- Line 224: The authors should define Health locus of control, health self-efficacy and patient activation.

In general, this is a well written paper with a well executed expirement. It was an interesting read.

---

## Round 0.2 · accepted · Accept

· Academic Editor

Accept

Looks good. Please just look at abstract --> Alivecor should be AliveCor. This is just a typo. You have this as AliveCore on p.7. Work with the production group to fix this error.